# Effectiveness of Negative Pulsed-Pressure Myofascial Vacuum Therapy and Therapeutic Exercise in Chronic Non-Specific Low Back Pain: A Single-Blind Randomized Controlled Trial

**DOI:** 10.3390/jcm11071984

**Published:** 2022-04-02

**Authors:** Manuel Rodríguez-Huguet, Jorge Góngora-Rodríguez, Maria Jesus Vinolo-Gil, Francisco Javier Martín-Vega, Rocío Martín-Valero, Daniel Rodríguez-Almagro

**Affiliations:** 1Department of Nursing and Physiotherapy, University of Cádiz, 11009 Cádiz, Spain; mariajesus.vinolo@uca.es (M.J.V.-G.); javier.martin@uca.es (F.J.M.-V.); 2Clinical Management Unit Rehabilitation, University Hospital of Puerta del Mar, 11009 Cádiz, Spain; 3Policlínica Santa María Clinic, 11008 Cádiz, Spain; jorgem.gongora@gmail.com or; 4University School of Osuna, University of Sevilla, 41640 Osuna, Spain; 5Department of Physiotherapy, Faculty of Health Sciences, University of Málaga, 29071 Málaga, Spain; rovalemas@uma.es; 6Department of Nursing, Physiotherapy and Medicine, University of Almería, 04120 Almería, Spain; dra243@ual.es

**Keywords:** negative pressure, vacuum, low back pain, cupping, physical therapy modalities

## Abstract

Non-specific low back pain is defined as pain located in the lumbar region; this condition is the most frequent musculoskeletal disorder. Negative pulsed-pressure myofascial vacuum therapy (vacuum treatment (VT)) devices mobilize tissue according to previously programmed parameters of force, time and frequency. The purpose of this study was to compare the effects of VT combined with core therapeutic exercise versus a physical therapy program (PTP) based only on core therapeutic exercise. Fifty participants with chronic non-specific low back pain were randomly assigned to two treatment groups, the VT group (*n* = 25) or the PTP group (*n* = 25). Pain, pressure-pain threshold, range of motion, functionality and quality of life were measured before treatment, at the end of treatment, and at one-month and three-month follow-ups. Both groups received 15 therapy sessions over 5 weeks. Statistically significant differences in favor of the VT group were shown in the results. In conclusion, the intervention based on myofascial vacuum therapy improved pain, mobility, pressure pain threshold, functionality and quality of life.

## 1. Introduction

Non-specific low back pain is defined as the presence of painful symptoms located in the lumbar region, which is between the lower margin of the ribs and the lower limit of the buttocks [1,2]. This symptomatology could condition movement due to fluctuations according to posture and physical activity, and it can be associated with radiating pain [1]. Conversely, the non-specific nature of this condition indicates that pain is not attributable to fractures, trauma or any other specific recognizable pathology (such as infectious, vascular or oncological conditions) [1,3]. Additionally, chronicity shows that symptoms have persisted for more than 12 weeks, as established by clinical practice guidelines for low back pain or lumbago [2,3,4,5].

Currently, low back pain (LBP) is the most frequent musculoskeletal pathology [6] and represents the main cause of disability worldwide, directly impacting individuals’ quality of life [7,8,9]. It represents a serious public health problem with increased socioeconomic repercussions due to its high prevalence, ranging between 22–65% [2]. It is widely estimated that up to 85% of people suffer episodes of LBP [7,10], and 20–25% of the population over 65 years of age suffers from chronic LBP [8]. Consequently, LBP patients suffer multiple limitations in carrying out basic daily tasks and activities, reducing their independence and affecting their professional activity and social life [10]. 

It is noteworthy that individual lifestyles may predispose the appearance of low back pain; sedentary behaviors are a risk factor in themselves [2] and are related to deficits in postural control and lower limb strength, which leads to LBP [6,10]. For this reason, avoiding bed rest is the fundament of non-specific chronic low back pain (CLBP) treatment, along with recommending the highest possible degree of activity, accompanied by educational measures and information on the pathology [1]. In this line, it has been demonstrated that physical exercise with therapeutic purpose, specifically lumbar stabilization and strengthening exercises, reduces pain and increases functionality and strength [8,9,11,12,13], maintaining these effects in the medium and long term [12]. 

Another aspect to consider is that within physical therapy treatment, it is possible to combine active treatment based on physical exercise with other techniques such as electrotherapy devices [4] or manual myofascial therapy techniques [14,15]. The latter includes several techniques based on applying negative pressure on the myofascial tissue, creating a negative pressure that results in the mobilization of myofascial tissue [16,17,18]. 

These techniques have important releasing effects on vessels and nerves, leading to improved vascularization, reduced neuropathic pain, normalization of myofascial tension, and, as a consequence of the above, a reduction of pain and an increase in mobility [19,20]. In spite of all previous benefits, the traditional application has the main limitation that it is not possible to quantify the applied pressure, hampering replication [21]. 

In order to address this problem, mechanical devices with digital parameterization have been developed in recent years. Negative pulsed-pressure myofascial vacuum therapy (Physium® Vacuum Treatment (VT)) is a pulsed negative-pressure therapy device that mobilizes tissue according to previously programmed parameters of force, time and frequency. It allows manual application by a therapist or static application by means of an articulated arm with a polypropylene treatment head at its end. Treatment head sizes vary from 20–100 cm^2^. Negative pressure can be preset from 25 to 250 millibars. The digital control of a wide range of parameters allows treatment to be tailored to each individual patient, thus avoiding the side effects of traditional cupping therapy [19,20,22], as well as the uncertainty in relation to the applied parameters.

The innovative character of this therapy, which could allow the mechanic stimulus applied to be controlled and quantified, might avoid the side effects of traditional cupping therapy, as well as the uncertainty in relation to the applied parameters. Thus, VT may lead to establishing a warranty and effective myofascial treatment. Due to the above, jointly with the novelty of the VT approach, the present study aims to compare the effect of a therapy program based on the joint approach of VT therapy and therapeutic physical exercise versus the isolated performance of the same therapeutic physical exercise programe.

## 2. Materials and Methods

### 2.1. Study Design

The design of the present study was a prospective and longitudinal single-blind randomized controlled clinical trial. It has been performed in accordance with the Helsinki Declaration and the CONSORT statement for reporting clinical trials. The trial has been registered in the Clinical Trials Registry with the reference number NCT04534179, and it has been approved by the Ethics Committee of Research of Cádiz (Spain) with the reference number 1095-N-20. 

### 2.2. Participants and Recruitment

The sample size calculation was carried out taking into account the data obtained from Lauche et al. [23], employing the software Epidat (Epidat: Epidemiological Analysis of Data, Version 3.1, January 2006, Consellería de Sanidade, Xunta de Galicia, Santiago de Compostela, Spain). Considering differences between groups in NPRS with a confidence level of 95% and a statistical power of 80%, the final sample size indicated that each group should rely on 25 patients. 

The criteria taken into account to include patients in the present study were: patients of both sexes, with an age range between 25 and 50 years old, diagnosed with CLBP (pain in the lumbosacral region, the persistence of pain for more than 12 weeks, pain not associated with specific systemic disease, and no signs of associated nerve irritation) [1,2]. On the other hand, pregnant patients, patients with pacemakers, those previously submitted to local surgery on the low back region, patients with fibromyalgia, lumbar radiculopathies or coagulopathies, cancer or other infectious processes were excluded from the present study.

All patients were recruited at the Santa María Clinic (Cádiz, Spain) by a clinician blinded as to group allocation. Each study group was composed through random allocation of the total sample with a 1:1 allocation ratio that was reported to patients by a different researcher, who was neither the one who performed the treatment nor the one who performed the evaluation, through an opaque envelope.

### 2.3. Measurements

The measurements of the study variables were made before the intervention, at the end of the intervention, and one month and three months after the conclusion of the treatment. Data were collected by an experienced physical therapist who was blinded as the evaluator.

Pain was the main study variable. This was measured with the Spanish version of the Numerical Pain Rate Scale (NPRS) [24]. This scale quantifies the intensity of pain between 0 (no pain) and 10 (worst pain) [25,26]. The NPRS has demonstrated, for musculoskeletal pain alterations, an intraclass correlation coefficient (ICC) between 0.61 and 0.95, a standard error of measurement (SEM) between 0.48 and 1.02, and a minimum detectable change (MDC) between 1.32 and 2.8 points [26,27,28]. 

The pressure-pain threshold (PPT) is the minimum applied force that induces pain. It is a validated method that presents reliability in the measurement with an ICC ranging from 0.78 to 0.93. The algometer consists of a disc of rubber with an area of 1 cm^2^ that is attached to a pressure pole. The measurements are expressed in kg/cm^2^, with the depression range from 0 to 10 kg/cm^2^. The PPT was measured bilaterally at the quadratus lumborum, piriformis, psoas and paravertebral muscles. The reliability and validity of such measurement in myofascial trigger points have been evaluated in numerous studies, determining that the algometer constitutes a reliable instrument for the quantification of pain sensitivity in patients with myofascial pain syndrome [29,30,31,32,33]. 

Lumbar spine mobility was evaluated with a digital inclinometer (Baseline^®^, Maharashtra, India). The parameters taken into account for this purpose were flexion, extension, lateroflexion and rotation movements. The physical therapist asked the patient to perform the aforementioned movements from a standing position, and the data were recorded at the end of the range of motion of each movement required [34].

The degree of functionality was registered through the Roland–Morris scale.The Roland–Morris Questionnaire is easy to apply, has high reliability and allows functionality to be determined, enabling the patient to be classified into different levels of disability [35,36,37,38].

Quality of life was assessed with the Spanish version of the SF-12 questionnaire. This instrument evaluates physical (PC-SF12) and mental (MC-SF12) factors of quality of life. In the Spanish population, the SF-12 questionnaire has demonstrated good reliability results (PC-SF12 Cronbach’s α = 0.85; MC-SF12 Cronbach’s α = 0.78) [39,40].

### 2.4. Procedures

The development of the intervention in both groups lasted five weeks. Individuals in the experimental group were subjected to a treatment of three weekly sessions of VT during the five weeks and a therapeutic exercise program (fifteen sessions in total), while the participants in the control group received the treatment of the therapeutic exercise program (physical therapy program (PTP)) with the same duration (fifteen sessions in total).

The treatment time of each VT session was 30 min. The applied pressure ranged between 80 and 100 mb in search of an analgesic effect, placing the applicators intermittently on the quadratus lumborum, piriformis, psoas and paravertebral muscles [19,20].

The therapeutic exercise program was based on core exercise [4,5,11,15,41]. The exercises were the same for both groups, although the control group only performed the indicated exercises, and the experimental group combined the exercises with VT sessions. The duration of the exercise protocol was five weeks, with three sessions per week, and an effective time of 20 min per session.

The exercises included in the treatment program were: supine bridge (pelvic lift from supine position), prone bridge (front plank), side bridge (lateral plank), dead bug (lifting one leg and the contralateral arm from a supine position) and bird dog (raising one leg and the contralateral arm from a quadrupedal position) [4,5,11,15,41]. All exercises of the therapeutic exercise program were performed by patients under the supervision of the physiotherapist (3 × 20 repetitions).

### 2.5. Statistical Analysis

In this study, continuous variables were described through means and standard deviations, and the description of the categorical variables was performed through frequencies and percentages. The normality of the distribution, as well as the equality of variances of the continuous variables, were tested employing the Kolmogorov–Smirnov test and Levene’s test, respectively. In order to determine the possible between-groups differences at baseline in relation to the morphologic characteristics and the descriptive data, Student´s t-test was used for continuous variables, and the chi-squared test was used for bivariate variables. The treatment effect was analyzed through a 2 × 4 mixed model of repeated measures analysis of variance, with time-by-group interaction as the main hypothesis. In the cases where differences between groups at baseline were appreciated, the analysis of the differences in change scores was controlled by the effect of the mentioned variable at baseline. The analysis of covariance (ANCOVA) was selected to perform the analysis in these specific cases. To test the differences between the VT and PTP groups through time (post-treatment, at 1 month and at 3 months), Student’s t-test for pre-change/post-change scores was applied. 

The effect sizes (ES) for time-by-group interactions were calculated using eta-squared (η^2^), which could be considered the analogue of R^2^, the determination coefficient employed for experimental studies. It can be interpreted as the percentage of the between-group differences because of the treatment effect. In addition, Cohen’s d was chosen to evaluate the ES in the bivariate analysis, and it was calculated as the mean of the between-groups difference divided by the pooled SD of both groups. Following Cohen’s recommendations [42], η^2^ was determined to be irrelevant when it was lower than 0.02, small when its value was between 0.02 and 0.15, medium when its value was between 0.15 and 0.35, and large when its value was over 0.35. The previous criteria also establish that a value of Cohen´s d < 0.2 could be considered irrelevant. Values of Cohen’s d between 0.2 and 0.49 were considered small, values between 0.5 and 0.8 were considered medium, and values >0.8 were considered large.

Previous clinical criteria [43,44] were employed to determine clinical success, where a reduction of 50% in NPRS values was considered a clinical success. Furthermore, in the present study, the number needed to treat (NNT) was calculated to reflect how many patients should be treated to obtain one more success or one less failure than would result if all patients were treated with the control treatment [45].

## 3. Results

At the end of the study, the entire sample had completely performed the treatment and all the evaluations required. Regarding the comparability of the morphologic characteristics of the sample at baseline, some variables revealed between-group differences. Nevertheless, in Table 1, it is possible to observe how, in all cases where between-group differences at baseline appear, VT group measurements seem to be worse than PTP group measurements. Descriptive data of the sample can be seen in Table 1.

Statistically significant differences in favor of the VT group were shown by the analysis of the variance. It also revealed an ES between medium and large, as well as a statistical power between 0.657 and 0.996. All the variables related to mobility showed a significant improvement, except for flexion movement (*p* = 0.087). For these variables, it was possible to observe medium ES values between 0.221 and 0.282. In addition, not only was an important enhancement in pain shown but this was also seen in algometry measures. All these results are shown in Table 2.

The result revealed statistically significant differences immediately post-treatment in the group treated with physical therapy (Table 3). Improvements were appreciated in all variables, except in the PPT of the left quadratus lumborum (mean difference =1.06; 95% CI = −10.02 to 12.44) and in the mental factor of quality of life (mean difference = 0.79; 95% CI = −2.72 to 4.29). The ES was large in most of the variables and ranged from 0.640 to 2.150 (Table 3). In post-treatment, the clinical success obtained in the group that received physical therapy was 92% (23 patients). Meanwhile, in the control group, the clinical success obtained after treatment was 72% (18 patients). The NNT was 5.00 (95% CI = 2.47 to −117.26).

At one month follow-up, the analysis also showed statistically significant differences in favor of the group treated with physical therapy (Table 4). A significant enhancement in pain, algometry, mobility and quality of life at the physical level was appreciated, with values of ES ranging between −0.596 and 1.943. At this time, changes in functionality were within the limit of signification (mean difference = −2.04; 95% CI = −4.08 to −0.001). At one month of follow-up, the clinical success obtained in the group that received physical therapy was 92% (23 patients). Meanwhile, in the control group, the clinical success obtained after treatment was 72% (18 patients). The NNT was 5.00 (95% CI = 2.47 to −117.26).

After three months of follow-up, the differences remained over time, as did the positive effect of therapy. Statistically significant differences in all variables relative to pain perception, pressure algometry, mobility, functionality and quality of life at the physical level were obtained in favor of the group treated with physical therapy (Table 5). After three months of follow-up, the clinical success obtained in the group that received physical therapy was 92% (23 patients). Meanwhile, in the control group, the clinical success obtained after treatment was 72% (18 patients). The NNT was 5.00 (95% CI = 2.47 to −117.26).

## 4. Discussion

The main findings derived from this study were the improvements obtained by the experimental treatment group based on decompressive myofascial therapy with pulsed negative pressure together with therapeutic exercise versus the control group, which consisted of only active exercise. The decompressive myofascial therapy and exercise treatment was comparatively more effective on pain, mobility, pressure pain threshold, functionality and quality of life levels than the exercise-only control group.

The achieved effects could be related to the changes induced by myofascial tissue mobilization when subjected to the negative pressure, resulting in increased blood flow to the area and modifications in the perception of the sensitivity of the area [16,19]. Thus the intervention modifies the state of central hypersensitization linked to the presence of proinflammatory cytokines suffered by individuals with chronic low back symptoms [7].

In the experimental treatment group, statistically significant differences were found between pre- and post-intervention measurements in all the variables analyzed. These changes were maintained over time, being observed in the long term: the effect continued to be reflected in the measurements taken one month and three months after the end of the treatments. These results correspond to those reported in previous articles on the subject, in which the effects derived from myofascial treatment with vacuum techniques, generally cupping, were analyzed [19,46]. 

Although existing studies are limited, there are only a few studies that compare a set of variables while waging on the combination of an instrumental treatment together with an active intervention based on exercise. These limitations are shown in previous studies that highlight the need to continue with studies that support the use of vacuum therapies [47]. 

With the proposed experimental intervention, a significant reduction in pain assessed by means of the NPRS is achieved, coinciding with the effects reported in the meta-analysis performed by Wang et al. (2017), where changes in the VAS scale are shown with therapeutic interventions with cupping versus other control techniques [46]. Normally, pain is listed as the main variable in studies analyzing low back pathology. Protocols proposed by Teut et al. (2018), Volpato et al. (2020) and Mardani et al. (2019) found positive, pain-reducing effects on the VAS scale, and Al Bedah et al. (2015) found pain-reducing effects on the NPRS [48,49,50,51].

In general, vacuum therapies are effective on lumbar symptoms, even when performed as a stand-alone treatment [49]. It is noteworthy that the effects are prolonged in time and maintained if long-term follow-up is carried out when the application of cupping is performed in a pulsed manner [48], which corroborates the results obtained with the treatment methodology applied in this project. The effects observed on the functionality assessed by means of the Roland-Morris Questionnaire should also be noted. This finding is especially relevant since functional capacity is especially limited in individuals with non-specific low back pain [52], such that the pulsed negative pressure treatment reaffirms the results of the analyzed literature, where disability is assessed by means of the aforementioned scales [49,50,51]. 

Regarding quality of life, improvements are obtained in the SF12 scale, implying that decompressive myofascial therapy with pulsed negative pressure contributes to improving the patient’s quality of life, as advocated in the study by Teut et al. (2018), which assessed quality of life by means of the extended version, the SF36 scale [48]. 

Similarly, the results coincide with research carried out using decompressive myofascial therapy in other locations of the spine, such as studies that applied these techniques in cervical pathology, finding pain reduction, improvements in mobility, decreased pressure pain threshold [22,53], and analgesic effects that stand out when the vacuum is applied in a pulsed manner [54]. Consequently, the changes observed in terms of lumbar mobility variables and pressure-pain threshold are in line with the previously cited findings on cervical pain when decompressive myofascial therapy is applied with pulsed negative pressure [53]. The results coincide with the clinical trial by Bonilla et al. (2015), with a common treatment basis in both cases [19], making the study particularly relevant because of the possibility of controlling the application parameters of the applied negative pressure as well as its pulsating capability. This condition prevents the appearance of reddening of the skin and hematomas, which otherwise occur in conventional cupping [18,22,23]. No patient suffered adverse effects derived from the intervention.

The fact that treatment parameters such as application time, pressure and vacuum intervals of the therapy can be controlled mechanically and digitally is a great advantage, representing the main strength of this research, since the intervention is reproducible, establishing general standardized action guidelines and making treatment conditions and parameters replicable. This possibility of quantifiable control of the intervention’s performance parameters is especially noteworthy, being the forte of our study since most of the research that develops treatments with vacuum therapies lack precise and quantifiable parameter controls [21]. On the other hand, it is possible to point out a limitation concerning the follow-up time, recommending a longer-term analysis for future projects. Likewise, it would be convenient to establish tools that allow the calculation of the optimal dose to be applied to each individual patient.

Therefore, in view of the results, it is possible to contemplate the combination of decompressive myofascial therapy with pulsed negative pressure with therapeutic physical exercise as a treatment of high value, capable of improving pain, mobility, pressure-pain threshold, functionality and quality of life in patients with non-specific chronic low back pain. Consequently, this combined treatment therapy may result in the reduction of socioeconomic costs derived from the medical care of individuals with chronic low back pain and could prevent and reverse the situation of overmedication that frequently accompanies these patients [55,56], being able to stand out as a preferential option in non-specific low back pain management [15]. Thus, the joint application of both therapies represents an interesting research line for continuing in future studies aiming at improving healthcare for the management of low back pain, with a commitment towards treatment models based on physical therapy.

In spite of the good results presented in this study, it is possible to find some limitations. It was not possible to include a placebo group, which would be advisable in order to evaluate the subjective influence of this treatment on patient pathology. Moreover, it was possible to observe a between-group difference in age. This aspect could be considered a limitation, but it is necessary to emphasize that the lifetime prevalence of low back pain is as high as 84%, depending on the case definition used, and no age group is spared, including children [57]. Furthermore, despite having a blinded evaluator, the patients were aware of their membership in each group, which could condition some aspects of the study, highlighting again the idea to carry out futures studies with a placebo group.

## 5. Conclusions

Myofascial vacuum therapy with pulsed negative pressure together with therapeutic exercise is an option in the treatment of chronic non-specific low back pain. The intervention based on myofascial vacuum therapy improves pain, mobility, pressure pain threshold, functionality and quality of life. 

## Figures and Tables

**Table 1 jcm-11-01984-t001:** Descriptive data and clinical characteristics of the sample at baseline.

Variable		Total (*n* = 50)		VT (*n* = 25)		PTP (*n* = 25)		*p*
		F	%	F	%	F	%	
Gender	Male	32		15		17		0.769
	Female	18		10		8		
**Continuous**		**Mean**	**SD**	**Mean**	**SD**	**Mean**	**SD**	
Age		37.18	10.82	43.32	8.49	31.04	9.40	<0.001
Weight (Kg)		75.75	15.17	78.29	16.64	73.32	13.49	0.256
Height (Cm)		171.2	0.09	1.70	0.08	172.2	0.09	0.391
NPRS		6.32	1.76	6.96	1.71	5.68	1.60	0.643
Flexion		36.84	12.63	36.00	13.69	37.68	11.69	0.016
Extension		15.22	5.28	13.48	4.90	16.96	4.95	0.003
Right Lateroflexion		20.24	6.31	17.68	5.86	22.80	5.78	0.002
Left Lateroflexion		20.80	6.38	18.08	6.00	23.52	5.62	0.178
Right Rotation		21.24	5.42	20.20	5.37	22.28	5.38	0.315
Left Rotation		21.36	5.29	20.60	5.27	22.12	5.31	0.009
PPT Right Quadratus Lumborum		3.89	0.76	3.65	0.81	4.14	0.64	0.023
PPT Left Quadratus Lumborum		3.83	0.81	3.54	0.89	4.13	0.59	0.010
PPT Right Piriformis		3.97	0.74	3.70	0.83	4.23	0.52	0.010
PPT Left Piriformis		4.01	0.72	3.72	0.82	4.29	0.46	0.004
PPT Right Psoas		3.68	0.82	3.56	0.89	3.80	0.73	0.306
PPT Left Psoas		3.74	0.72	3.66	0.77	3.82	0.68	0.449
PPT Right Paravertebral		4.03	0.73	3.68	0.77	4.38	0.50	<0.001
PPT Left Paravertebral		4.05	0.67	3.72	0.68	4.37	0.49	<0.001
Roland–Morris		6.78	4.48	7.72	4.59	5.84	4.25	0.139
SF12 Physical Factor		43.59	9.71	41.24	7.84	45.30	10.65	0.137
SF12 Mental Factor		52.85	9.89	51.30	9.04	53.75	8.69	0.340

Abbreviations. SD: standard deviation; VT: vacuum treatment; PTP: physical therapy program; PPT: pressure pain threshold; NPRS: Numerical Pain Rating Scale; *p*: *p*-value.

**Table 2 jcm-11-01984-t002:** Statistical significance, effect size and power of the time-by-group interaction from the analysis of variance.

Variable	F	*p*	Effect Size (η^2^)	Power
NPRS	4.866	0.005	0.245	0.881
Flexion	2.323	0.087	0.132	0.548
Extension	5.894	0.002	0.282	0.937
Right Lateroflexion	5.743	0.002	0.277	0.931
Left Lateroflexion	5.136	0.004	0.255	0.899
Right Rotation	4.353	0.009	0.221	0.840
Left Rotation	4.711	0.006	0.235	0.870
PPT Right Quadratus Lumborum	6.571	0..001	0.305	0.960
PPT Left Quadratus Lumborum	6.540	0.001	0.304	0.956
PPT Right Piriformis	3.506	0.021	0.192	0.753
PPT Left Piriformis	2.928	0.044	0.163	0.657
PPT Right Psoas	6.463	0.001	0.297	0.957
PPT Left Psoas	9.714	<0.001	0.388	0.996
PPT Right Paravertebral Muscles	7.212	<0.001	0.325	0.974
PPT Left Paravertebral Muscles	6.781	0.001	0.311	0.965
Roland–Morris	2.501	0.071	0.140	0.582
SF12 Physical Factor	2.892	0.046	0.168	0.649
SF12 Mental Factor	0.597	0.621	0.040	0.164

Abbreviations. *p*: *p*-value; PPT: pressure-pain threshold; NPRS: Numerical Pain Rating Scale.

**Table 3 jcm-11-01984-t003:** Within-groups and between-groups differences in post-treatment.

Variable		Post-Treatment		Within-Group Change Score		Between-Groups Change Score		Effect Size	
		Mean	SD	Mean	SD	Mean Difference (95% CI)	*p*	D	ES
NPRS	VT	1.2	1.83	9.16	9.2	−2.36 (−3.37 to −1.35)	<0.001	−0.721	Medium
	PTP	2.28	1.9	3.68	5.57				
Flexion	VT	45.16	7.65	3.24	3.83	5.48 (1.11 to 9.85)	0.015	0.760	Medium
	PTP	41.36	10.59	0.68	2.83				
Extension	VT	18.72	4.03	8.36	3.89	4.56 (2.642 to 6.48)	<0.001	1.471	Large
	PTP	17.64	5.74	2.76	3.72				
Right Lateroflexion	VT	26.04	4.03	7.56	3.94	5.60 (3.43 to 7.77)	<0.001	1.378	Large
	PTP	25.56	4.11	2.16	3.9				
Left Lateroflexion	VT	25.64	4.41	6.04	4.46	3.20 (0.74 to 5.66)	<0.001	0.872	Large
	PTP	24.68	4.01	2.32	4.06				
Right Rotation	VT	26.24	4.39	5.64	4.65	3.72 (1.30 to 6.14)	0.003	0.739	Medium
	PTP	24.6	4.72	2.44	3.98				
Left Rotation	VT	26.24	4.41	−3.76	2.01	3.20 (0.74 to 5.66)	0.012	0.201	Small
	PTP	24.56	4.74	−3.40	1.53				
PPT Right Quadratus Lumborum	VT	4.69	0.67	1.04	0.72	0.80 (0.47 to 1.13)	<0.001	1299	Large
	PTP	4.38	0.58	0.24	0.49				
PPT Left Quadratus Lumborum	VT	4.70	0.44	18.62	20.32	1.06 (−10.02 to 12.44)	0.848	0.055	Irrelevant
	PTP	4.40	0.60	17.55	18.61				
PPT Right Piriformis	VT	4.71	0.53	1.00	0.83	0.788 (0.43 to 1.15)	<0.001	1.224	Large
	PTP	4.45	0.58	0.22	0.35				
PPT Left Piriformis	VT	4.72	0.51	0.99	0.82	0.84 (0.48 to 1.19)	<0.001	1.334	Large
	PTP	4.45	0.53	0.16	0.32				
PPT Right Psoas	VT	4.55	0.79	0.99	0.69	0.87 (0.57 to 1.18)	<0.001	1.644	Large
	PTP	3.91	0.78	0.12	0.29				
PPT Left Psoas	VT	4.55	0.81	0.88	0.48	0.82 (0.60 to 1.05)	<0.001	2.150	Large
	PTP	3.88	0.8	0.05	0.26				
PPT Right Paravertebral	VT	4.76	0.34	1.08	0.67	0.91 (0.60 to 1.21)	<0.001	1.712	Large
	PTP	4.55	0.54	0.17	0.34				
PPT Left Paravertebral	VT	4.77	0.33	4.69	0.47	0.13 (0.59 to 1.12)	<0.001	0.587	Medium
	PTP	4.56	0.51	4.38	0.58				
Roland–Morris	VT	1.92	3.04	−5.80	3.65	−2.28 (−4.30 to −0.26)	0.028	−0.640	Medium
	PTP	2.32	3.47	−3.52	3.47				
SF12 Physical Factor	VT	49.04	8.99	7.80	7.79	1.95 (0.99 to 8.87)	0.015	0.640	Medium
	PTP	48.17	9.49	2.87	5.77				
SF12 Mental Factor	VT	52.94	7.44	1.64	7.35	0.79 (−2.72 to 4.29)	0.651	0.129	Irrelevant
	PTP	54.59	7.34	0.85	4.51				

Abbreviations. SD: standard deviation; D: Cohens´s d; ES: effect size; *p*: *p*-value; VT: vacuum treatment; PTP: physical therapy program; PPT: pressure-pain threshold; NPRS: Numerical Pain Rating Scale.

**Table 4 jcm-11-01984-t004:** Within-groups and between-groups differences at 1-month follow-up.

Variable		1-Month Follow-Up		Within-Group Change Score		Between-Groups Change Score		Effect Size	
		Mean	SD	Mean	SD	Mean Difference (95% CI)	*p*	D	ES
NPRS	VT	0.68	1.49	−6.28	1.81	−2.64 (−3.70 to −1.58)	<0.001	−1.419	Large
	PTP	2.04	2.11	−3.64	1.91				
Flexion	VT	44.92	6.48	8.92	9.62	5.00 (0.49 to 9.50)	0.031	0.634	Medium
	PTP	41.60	10.38	3.92	5.62				
Extension	VT	18.16	4.25	4.68	3.76	4.28 (2.40 to 6.16)	<0.001	1.296	Large
	PTP	17.36	6.04	0.40	2.77				
Right Lateroflexion	VT	26.28	4.43	8.60	4.39	6.36 (4.02 to 8.70)	<0.001	1.547	Large
	PTP	25.04	4.35	2.24	3.81				
Left Lateroflexion	VT	25.80	4.56	7.72	4.31	6.00 (3.69 to 8.31)	<0.001	1.477	Large
	PTP	25.24	4.16	1.72	3.80				
Right Rotation	VT	27.00	3.58	6.80	5.08	4.60 (2.06 to 7.15)	0.001	1.030	Large
	PTP	24.48	4.58	2.20	3.75				
Left Rotation	VT	27.12	3.56	6.52	4.76	4.20 (1.78 to 6.62)	0.001	0.986	Large
	PTP	24.44	4.60	2.32	3.69				
PPT Right Quadratus Lumborum	VT	4.70	0.47	1.05	0.69	0.78 (0.46 to 1.11)	<0.001	1.401	Large
	PTP	4.40	0.56	0.26	0.40				
PPT Left Quadratus Lumborum	VT	4.71	0.46	1.16	0.78	0.90 (0.55 to 1.25)	<0.001	1.474	Large
	PTP	4.39	0.60	0.26	0.37				
PPT Right Piriformis	VT	0.318	0.318	0.94	0.83	0.71 (0.35 to 1.07)	<0.001	1.125	Large
	PTP	0.318	0.318	0.22	0.36				
PPT Left Piriformis	VT	4.73	0.50	1.01	0.79	0.84 (0.50 to 1.18)	<0.001	1.394	Large
	PTP	4.46	0.54	0.17	0.32				
PPT Right Psoas	VT	4.57	0.78	1.01	0.64	0.93 (0.65 to 1.20)	<0.001	1.943	Large
	PTP	3.88	0.78	0.08	0.22				
PPT Left Psoas	VT	4.55	0.77	0.88	0.55	0.81 (0.56 to 1.06)	<0.001	1.846	Large
	PTP	3.90	0.79	0.08	0.27				
PPT Right Paravertebral	VT	4.79	0.34	1.08	0.64	0.89 (0.61 to 1.18)	<0.001	1.769	Large
	PTP	4.57	0.49	0.19	0.31				
PPT Left Paravertebral	VT	4.78	0.32	1.06	0.62	0.86 (0.59 to 1.14)	<0.001	1.763	Large
	PTP	4.57	0.50	0.19	0.32				
Roland–Morris	VT	1.48	2.62	−6.24	3.91	−2.04 (−4.08 to −0.001)	0.050	−0.569	Medium
	PTP	1.64	2.89	−4.20	3.23				
SF12 Physical Factor	VT	49.32	8.59	8.41	7.18	5.28 (1.36 to 9.21)	0.009	0.791	Medium
	PTP	48.42	9.39	3.12	6.16				
SF12 Mental Factor	VT	53.08	5.90	2.28	6.72	1.55 (-1.86 to 4.96)	0.364	0.266	Small
	PTP	54.48	7.02	0.73	4.75				

Abbreviations. SD: standard deviation; D: Cohens´s d; ES: effect size; *p*: *p*-value; VT: vacuum treatment; PTP: physical therapy program; PPT: pressure-pain threshold; NPRS: Numerical Pain Rating Scale.

**Table 5 jcm-11-01984-t005:** Within-group and between-group differences at 3 months follow-up.

Variable		3-Month Follow-Up		Within-Group Change Score		Between-Groups Change Score		Effect Size	
		Mean	SD	Mean	SD	Mean Difference (95% CI)	*p*	D	Effect
NPRS	VT	0.76	1.48	−6.20	1.78	−2.52 (−3.56 to −1.48)	<0.001	−1.384	Large
	PTP	2.00	2.06	−3.68	1.86				
Flexion	VT	45.08	6.92	9.08	9.34	5.24 (0.84 to 9.64)	0.021	0.677	Medium
	PTP	41.52	10.42	3.84	5.70				
Extension	VT	18.28	3.90	4.80	3.60	4.56 (2.74 to 6.38)	<0.001	1.427	Large
	PTP	17.20	5.97	0.24	2.73				
Right Lateroflexion	VT	26.24	4.18	8.56	4.41	6.08 (3.78 to 8.38)	<0.001	1.504	Large
	PTP	25.28	4.36	2.48	3.64				
Left Lateroflexion	VT	26.08	4.19	8.00	4.22	6.2 (3.94 to 8.46)	<0.001	1.560	Large
	PTP	25.32	4.20	1.80	3.71				
Right Rotation	VT	27.12	3.63	6.92	5.00	4.64 (2.11 to 7.17)	0.001	1.041	Large
	PTP	24.56	4.51	2.28	3.84				
Left Rotation	VT	27.16	3.59	6.56	4.74	2.24 (1.22 to 6.66)	0.001	0.998	Large
	PTP	24.44	4.60	2.32	3.69				
PPT Right Quadratus Lumborum	VT	4.71	0.49	1.06	0.15	0.79 (0.44 to 1.14)	<0.001	6.572	Large
	PTP	4.40	0.57	0.27	0.08				
PPT Left Quadratus Lumborum	VT	4.76	0.45	1.21	0.16	0.95 (0.58 to 1.31)	<0.001	7.693	Large
	PTP	4.39	0.59	0.26	0.07				
PPT Right Piriformis	VT	4.72	0.54	1.02	0.16	0.77 (0.40 to 1.14)	<0.001	6.166	Large
	PTP	4.48	0.58	0.24	0.08				
PPT Left Piriformis	VT	4.73	0.51	1.01	0.16	0.79 (0.44 to 1.14)	<0.001	6.397	Large
	PTP	4.51	0.58	0.22	0.07				
PPT Right Psoas	VT	4.58	0.78	1.02	0.13	0.93 (0.64 to 1.23)	<0.001	9.544	Large
	PTP	3.88	0.53	0.08	0.05				
PPT Left Psoas	VT	4.56	0.78	0.89	0.11	0.82 (0.58 to 1.06)	<0.001	9.597	Large
	PTP	3.89	0.77	0.07	0.05				
PPT Right Paravertebral	VT	4.80	0.30	1.12	0.13	0.91 (0.61 to 1.21)	<0.001	9.087	Large
	PTP	4.58	0.52	0.2	0.06				
PPT Left Paravertebral	VT	4.80	0.31	1.08	0.12	0.88 (0.59 to 1.16)	<0.001	9.060	Large
	PTP	4.57	0.51	0.19	0.07				
Roland Morris	VT	1.40	2.50	−6.32	3.82	−2.24 (−4.26 to −0.22)	0.031	−0.629	Medium
	PTP	1.76	2.96	−4.08	3.28				
SF12 Physical Factor	VT	49.57	8.41	8.33	7.34	5.24 (1.34 to 9.14)	0.009	0.777	Medium
	PTP	48.39	9.20	3.09	6.09				
SF12 Mental Factor	VT	53.27	5.97	1.97	6.53	1.17 (−2.13 to 4.47)	0.479	−0.195	Irrelevant
	PTP	54.54	6.99	0.80	4.77				

Abbreviations. SD: standard deviation; D: Cohens’s d; ES: effect size; *p*: *p*-value; VT: vacuum treatment; PTP: physical therapy program; PPT: pressure-pain threshold; NPRS: Numerical Pain Rating Scale.

## Data Availability

The trial was registered in the Clinical Trials Registry (reference number NCT04534179).

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
