# Peer review of "Effectiveness of Negative Pulsed-Pressure Myofascial Vacuum Therapy and Therapeutic Exercise in Chronic Non-Specific Low Back Pain: A Single-Blind Randomized Controlled Trial"

_jcm, 2022, doi:10.3390/jcm11071984_

Round 1
Reviewer 1 Report
- Negative Pulsed Pressure Myofascial Vacuum Therapy was treated on the quadratus lumborum, piriformis, psoas and paravertebral muscles. The authors should use photos to illustrate.
- Age comparisons between VT group(43.32±8.49 years) and PTP group (31.04±9.40) are statistically different(P<0.001) at baseline. This age descriptive data may affect the validity of experimental results.
- Data were collected by an experienced physical therapist who was blinded as evaluator. However, subjects would know how they are being treated with vacuum therapy or physical therapy program. It is possible for the evaluator to know from the patient to which group the patient belongs. The authors should mention some limitations of this study in the discussion.
- Gender data did not presented in Table 1. Descriptive data and clinical characteristics of the sample at baseline.
Author Response
Respuesta al revisor 1 Comentarios.
Punto 1. La terapia de vacío miofascial con presión negativa pulsada se trató en los músculos cuadrado lumbar, piriforme, psoas y paravertebral. Los autores deben usar fotos para ilustrar.
Respuesta 1. Se han incluido imágenes de las diferentes estructuras tratadas con Terapia de Vacío Miofascial de Presión Pulsada Negativa. Muchas gracias por mejorar nuestro manuscrito.
Punto 2. Las comparaciones de edad entre el grupo VT (43,32±8,49 años) y el grupo PTP (31,04±9,40) son estadísticamente diferentes (P<0,001) al inicio del estudio. Estos datos descriptivos de edad pueden afectar la validez de los resultados experimentales.
Response 2. Thanks for the careful review. A great set of factors are related with low back pain occurrence, of which age is not one of the most relevant. Furthermore, lifetime prevalence of low back pain is as high as 84%, depending on the case definition used, and no age group is spared, including children. In addition, this issue has been mentioned in discussion section as a possible study limitation, between lines 341 and 344.
Point 3. Data were collected by an experienced physical therapist who was blinded as evaluator. However, subjects would know how they are being treated with vacuum therapy or physical therapy program. It is possible for the evaluator to know from the patient to which group the patient belongs. The authors should mention some limitations of this study in the discussion.
Respuesta 3. El evaluador fue un investigador diferente del que realiza la asignación y del que realiza la intervención, cegado a la asignación de pacientes. Para que nuestro manuscrito sea más fácil de entender, este problema se ha modificado en la sección de métodos entre las líneas 107 y 111. Además, esta preocupación se ha incluido en la sección de discusión como una posible limitación del estudio, entre las líneas 345 y 347. Gracias por su cuidadoso revisión.
Punto 4. Datos de género no presentados en la Tabla 1. Datos descriptivos y características clínicas de la muestra al inicio del estudio.
Respuesta 4. Estamos muy agradecidos con las mejoras propuestas por el revisor. Los datos de género se han agregado en la tabla 1.

Reviewer 2 Report
There is 20% repetition rate in the study excluding references and quotations. Methods and statistical analysis are highly similar. High repetition rate is not acceptable.
A lot of paragraphs most of them single sentences without any coherence are written in the introduction.
Line 45, line 59 -62, 63-68, 69-71, 81- 84 – Why are these sentences required here?
The vacuum therapy is not properly introduced in the introduction section
The introduction has to be improved adding rationale for this study
What is the aim of this study? It should be there in the introduction
Which version of the NPRS and SF12 was used in this study? Spanish or English
How was the lumbar ROM measured? This has to be explained
Why Kolmogorov test is used for normality testing in small sample size?
What were the components of exercise? Why these exercises were selected? Are these the best exercise for LBP?
The discussion has a lot of paragraphs with single sentences. Ingle sentences cannot form a paragraph
Add limitation and future research to the discussion
Author Response
Response To Reviewer 2 Coments.
Point 1. There is 20% repetition rate in the study excluding references and quotations. Methods and statistical analysis are highly similar. High repetition rate is not acceptable.
Response 1. Thanks for your careful review. Attending the methods section, especially statistical analysis subsection, has been modified with the purpose of avoid the high repetition rate. These changes have been performed between lines 87-101, 104-111, 118-121, 131-135, 139-142, 164-187 and 191-194. Thank you very much for make better our manuscript.
Point 2. A lot of paragraphs most of them single sentences without any coherence are written in the introduction. Line 45, line 59 -62, 63-68, 69-71, 81- 84 – Why are these sentences required here?
Response 2. In order to improve the quality of our manuscript, and in an attempt to answer the 2, 3 and 4 recommendations suggested by the reviewer, introduction section has been rewritten. The changes have been carried out between lines 32-40, 49-70 and 76-84. We are very thankful with the improvements proposed by the reviewer.
Point 3. The vacuum therapy is not properly introduced in the introduction section
Response 3. Thanks to the reviewer for this concern. Introduction section has been modified with the objective of make our manuscript more comprehensive. In this aspect, vacuum therapy has been introduced between lines 60 and 77.
Point 4. The introduction has to be improved adding rationale for this study. What is the aim of this study? It should be there in the introduction
Response 4. Following the reviewer recommendation, the article has been duly justified and the objective has been added to introduction section between lines 78 and 84. Thank you for your careful review.
Point 5. Which version of the NPRS and SF12 was used in this study? Spanish or English
Response 5. Following the reviewer advice, we have added the version of the questionnaires employed, in lines 116 and 139. In addition, as consequence of your appreciation we have find a mistake with the reference employed for NPRS questionnaire, that it has been already solved introducing the correct reference. Thank you very much.
Point 6. How was the lumbar ROM measured? This has to be explained
Response 6. Thanks very much to the reviewer for the advice. An explanation of the lumbar ROM procedure has been included between lines 131 and 135.
Point 7. Why Kolmogorov test is used for normality testing in small sample size?
Response 7. We are so grateful with your appreciation. This is a concern that we took into account at the beginning because Saphiro-Wilk test was originally restricted for sample sizes of less than 50 subjects. In our study, due to our sample size is in the limit, we decided that the best way to test normality was to use Kolmogorov-Smirnov test, that is recommended when the sample size is over 50 subjects.
Razali, Nornadiah Mohd, and Yap Bee Wah. Power comparisons of shapiro-wilk, kolmogorov-smirnov, lilliefors and anderson-darling tests." Journal of statistical modeling and analytics 2.1 (2011): 21-33.
Point 8. What were the components of exercise? Why these exercises were selected? Are these the best exercise for LBP?
Response 8. Thanks to the reviewer for this concern In this concern, the exercises performed in the present study are explained in methods section, procedures subsection, between lines 157 and 162. These exercises have been selected taking into account the apported references in lines 152 and 160, more specifically references number 4,5,11,15 and 41.
Point 9. The discussion has a lot of paragraphs with single sentences. Ingle sentences cannot form a paragraph
Response 9. Thank you very much for your appreciation. Following your advice, discussion section has been modified respecting the previous content.
Point 10. Add limitation and future research to the discussion.
Response 10. Thanks to the reviewer for this observation that make better our manuscript. It has been added new text, that make reference to the study limitations, in discussion section between lines 339 and 347.
